# A game theoretic treatment of contagion in trade networks

**John S. McAlister**[1,2,3]*, **Jesse L. Brunner**[4], **Danielle J. Galvin**[5], **Nina H. Fefferman**[2,3,6]

**1** Department of Mathematics, University of Tennessee - Knoxville, Knoxville, Tennessee, United States of America, **2** National Institute for Modeling Biological Systems, Knoxville, Tennessee, United States of America, **3** NSF Center for Analysis and Prediction of Pandemic Expansion, Phoenix, Arizona, United States of America, **4** School of Biological Sciences, Washington State University, Pullman, Washington, United States of America, **5** Center for Wildlife Health, School of Natural Resources, University of Tennessee - Knoxville, Knoxville, Tennessee, United States of America, **6** School of Mathematical and Natural Sciences, Arizona State University, Phoenix, Arizona, United States of America

* jmcalis6@vols.utk.edu

## Abstract

Global trade of material goods involves the potential to create pathways for the spread of infectious pathogens. One trade sector in which this synergy is clearly critical is that of wildlife trade networks. This highly complex system involves important and understudied bidirectional coupling between the economic decision making of the stakeholders and the contagion dynamics on the emergent trade network. While each of these components are independently well studied, there is a meaningful gap in understanding the feedback dynamics that can arise between them. In the present study, we describe a general game theoretic model for trade networks of goods susceptible to contagion. The primary result relies on the acyclic nature of the trade network and shows that, through the course of trading with stochastic infections, the probability of infection converges to a directly computable fixed point. This allows us to compute best responses and thus identify equilibria in the game. We present ways to use this model to describe and evaluate trade networks in terms of global and individual risk of infection under a wide variety of structural or individual modifications to the trade network. In capturing the bidirectional coupling of the system, we provide critical insight into the global and individual drivers and consequences for risks of infection inherent in and arising from the global wildlife trade, and any economic trade network with associated contagion risks.

## Author summary

When networks of stakeholders trade goods that can become contaminated with an infection, like animal diseases in the pet trade or wood pests in the lumber trade, there is a trade off between minimizing cost and maximizing health

**Data availability statement:** All the data in this manuscript was synthetically generated by the model using the code found in the repository https://github.com/feffermanlab/JSM_2024_WildlifeTradeNetworks.

**Funding:** NHF and JLB received funding for this work from the National Science Foundation Division of Environmental Biology (NSF DEB#2207922 https://www.nsf.gov/awardsearch/showAward?AWD_ID=2207922&HistoricalAwards=false). Additionally, NHF received funding for this work from the National Science Foundation Division of Biological Infrastructure (NSF DBI #2412115 https://www.nsf.gov/awardsearch/showAward?AWD_ID=2412115&HistoricalAwards=false). The funders had no role in study design, data collection and analysis, decision to publish, or preparation of the manuscript.

**Competing interests:** The authors have declared that no competing interests exist.

and safety. The most efficient choice for each stakeholder is determined by the entire trade network of which they are a part. This paper introduces a model that can be used to understand the relationship between the structured of the network and the individual outcomes influenced by economic and ecological feedback loops throughout the trade network.

## 1 Introduction

Understanding infection and contagion in trade networks combines several important areas of study. The economic decision-making involved in the actions of trading and investing in measures to prevent infection, the contagion dynamics on a network, and the risk of spillover from the closed network into the environment, have each been studied on their own. All three elements separately are useful for informing bits and pieces of a system wherein many stakeholders are trading products which may transmit some infection. However, without considering each of the elements together, we cannot understand the bidirectional coupling between the health and safety practices and the economic decision making. Stakeholders make decisions about health and safety measures based on infection dynamics and perceived risk, the infection dynamics are changed by these decision which are also constrained by economic factors, and the structure of the network by which the players are connected may have a great impact on this interaction. Here we present a first model which integrates all of these components using infectious disease in wildlife trade networks (e.g., for pets) as a motivating example. Because the wildlife trade is highly studied and because the wildlife trade gives an easy to imagine example of a product being traded that can transmit a contagion, most of the work described in this manuscript is framed around wildlife trade as a model system. However, any system wherein the products being traded may be infected with a contagion (e.g., mold in wooden products [1], or the transmission of tree pests in fire wood trade [2]) can be examined by this model equally well.

Much of the work on the risks that result from wildlife trade has focused either on conservation or invasion biology [3–5]. For instance, several models have focused on the trade of agricultural or hunted animals (e.g., [6,7]), statistically estimating risks based on the ecological niche of relevant species and habitat characteristics necessary for exposure. While invasion from exotic wildlife being traded poses a large threat to ecosystems, it is not the only threat posed by such trade. Spillover of wildlife diseases poses a large threat as well but has thus far primarily been examined either statistically (looking for patterns in observed data) [8–10] or theoretically at an international scale, without regard for the economic incentives and behaviors of the trade network participants. Moreover, most of the focus has been on the spillover risk specifically to humans, e.g., [11–13], rather than also considering the impact to native wildlife or local agricultural populations. This is not to say we are alone; some studies have begun to consider these questions, examining the underlying economic incentive structures for human-environmental interactions, e.g., [14]. Some very elegant

work has also explored the explicit economics of the interaction between animal trade and infection, but absent the framework of a trade network (e.g., [15]).

In addition to the work done on invasion and spillover resulting from wildlife trade and other similar industries, there is a large body of work regarding the game theoretic elements of trade [16–18]. Understanding economic behavior, even in complex networks is not the novel element of this work. Rather we propose this model as a unique connection between the economic view of trade networks and biological view of spillover risk.

Separate from both of these things, there is a rich literature about the passage of a contagion or infection through networks (e.g., [19–22] and many others). Investigations of infection through networks have existed since the beginning of network science and has been used for applications ranging from understanding social strategies for viral marketing campaigns [23] to predicting spatio-temporal patterns in mortgage default behavior [24]. However, even when the economic and biological components have been studied together, to the best of our knowledge, there has been no work on the impact that the explicit network structure has on the coupled feedback between economic decision-making and infection risk in trade networks themselves which is a crucial piece of designing better, more resilient trade networks.

Each of these types of models, on its own, is not sufficient to understand bidirectional coupling between health and safety practices and economic decision making. When stakeholders are making decisions about health and safety measures based on economic incentives, which change based on the perceived risk, and the perceived risk changes in response to changes in health and safety decisions, we see a new set of rich dynamics worth investigating. The key reason to understand this coupling is to advance our capabilities in predictive modeling towards making informed decisions about spillover risk from trade networks of products vulnerable to contagion. In the present study, we build a game theoretic model which is flexible and robust, and that captures the relationship between investment in health and safety practices and interactions within the trade network. The model presented here is not predictive but provides a framework for understanding these kinds of systems with more accurate insight. This model is usable because, as we show, there exists a long-time stable infection probability. This insight allows us to treat a typically stochastic system deterministically and compute best responses. Examining this system by studying best responses (and thus Nash equilibria) allows us to find points in the strategy space where players, making rational decisions to maximize their own payoff, are balancing the cost of health and safety practices with the benefit it gives them in the trade network.

In Sect 2.1, we describe the general model and the crucial hypotheses that allow it to be usable. In Sect 2.2, we present the result about the existence of the directly computable stationary probability of infection and describe the process to derive it. Equipped with this result we discuss the existence of best response functions in Sect 3 and provide an easy set of sufficient conditions for the existence of such a function. In Sect 4, we fill in the general functions of the model with explicit functional forms that give us a toy model to examine numerically. We discuss the results from the toy model and its implications for the application areas of this model in Sect 5. Readers interested primarily in the utility and application of this game-theoretic model may wish to skip to the toy model and discussion.

## 2 Methods

### 2.1 Trade network contagion

Consider a group of $n$ players each with a strategy $x_i \in \mathbb{R}^s$ and with a contagion status $I_i \in \{0, 1\}$ (i.e., if a player $i$ is infected then $I_i = 1$ otherwise $I_i = 0$). The strategy $x_i$ may represent a combination of many elements of trade strategy like price, investment in health and safety, surveillance up stream, etc. The contagion status is simply the presence or absence of the infection in the stock held by a particular player. Let the collection of all $n$ strategies be called $X := [x_1, x_2, ..., x_n] \in \mathbb{R}^{s \times n}$. Furthermore, let the collection of all strategies except for that of player $i$ be called $X_{-i} \in \mathbb{R}^{s \times n-1}$ and let $I$ be the vector of contagion states $[I_1, I_2, ..., I_n]^T$ so that $(X, I)$ represents present state of the game as a whole. The game considers these $n$ players interacting with one another in an acyclic trade network, passing goods downstream through the trade network at rates which depend on the strategies of all players in the network.

Our addition to the game is to consider how the presence of a contagion, which may transmitted by the goods being traded, changes the dynamics of the game. We consider only acyclic trade networks which means that contagion can only be passed in one direction, "downstream," through the network, after it is introduced with probability $\epsilon_i$ from the environment to player $i$. This assumes a dynamic in which infections risks are associated with the transfer of traded products themselves, rather than by mere contact between participants. To that end, we give the players an order so that if $i$ and $j$ interact and $i$ is contagious, $j$ may become contagious if and only if $i < j$. This is equivalent to giving the vertices in the directed graph representation of the trade network a topological ordering which necessarily exists because the graph is acyclic. The probability that a focal individual $i$ interacts with player $j < i$ (upstream) at a particular time is given by $a_{i,j}(X, I_j)$. That is, the probability of interaction depends on both the strategies and contagion state of player $j$. Such an interaction can be thought of as a player $i$ purchasing from player $j$. We say that the cost to the focal player $i$ is given as $c_{i,j}(x_i, x_j)$ and from the perspective of the upstream player, $j$, (i.e., the seller) the benefit is given by $b_{j,i}(x_i, x_j)$. Lastly, let $d_i(x_i, I_i)$ be the component of payoff which does not depend on interactions with other players in the network. It is most natural to think of $b_{i,j}$ as the price at which $i$ sells a good to $j$, $c_{i,j}$ as the cost of $i$ buying a good from $j$, and $d_i$ as the amount that player $i$ "enjoys" owning the good given their strategy and contagion state. Through all of this, we get the following payoff function, $\pi_i$.

$$\pi_i(x_i, X_{-i}, I) = d_i(x_i, I_i) + \sum_{j=1}^{i-1} c_{i,j}(x_i, x_j)a_{i,j}(X, I_j) + \sum_{j=i+1}^{n} b_{i,j}(x_i, x_j)a_{j,i}(X, I_i). \tag{1}$$

We now express these as matrices and we get $A(X, I)$, a strictly lower triangular matrix with rows $\|\vec{a}\|_{L^1} \leq 1$; $B$, an upper triangular matrix; and $C$ a lower triangular matrix. Further, let $F$ be component wise multiplication of $C$ and $A$, and let $G$ be the component wise multiplication of $B$ and $A^T$. Let $d = [d]_{i=1}^n$ and we get

$$\pi(X, I) = d(X, I) + (F(X, I) + G(X, I))\mathbf{1}$$

where $\mathbf{1}$ is the vector of all 1s, $I$ is the vector of contagion states, and $X$ is the strategy profile across all players (an $n \times m$ matrix).

Underlying this system is a system of contagion where each player, $x_i$, with a contagion state $I_i = 0$ may become infected with some probability, $\alpha_i(x_i)$, by interacting with an infected player, or, with probability $\beta_i(x_i)$, from an environmental source. We assume that if a contagion is introduced to the stock of a stakeholder, their entire stock can be considered infectious. For this reason, this model is more appropriate to modeling fast spreading contagions. However, the results from this model still give an absolute upper bound for contagion presence in the case of incomplete spread of an infection. Infected players (i.e., $x_j$ with contagion state $I_j = 1$) may recover with probability $f(x_i)$. The transitions between health states are shown in Fig 1.

We seek to solve this game by finding *long time Nash equilibria*, which are states when every player plays a strategy which, when played for a long time, gives an average payoff that cannot be improved upon unilaterally by that player [25]. In other words, each player is playing their best response (in a long term average sense) to their co-players.

## 2.2 Stationary probabilities of infection

To seek long time Nash equilibria, we first must find the expected value of being infected at any time. Here we present a result which says that, regardless of the initial contagion state of the system, if contagion is introduced by environmental factors at a low amount and percolated through a network downstream in the fashion described above, the probability of being infected at any given time will converge to a stationary probability vector, so long as the graph is finite. The crucial element is to note that, because we can give the vertices an ordering so that the contagion is only passed from smaller vertices to larger vertices, this graph, as a digraph of contagion, is acyclic (it can be given a topological ordering) (Fig 2).

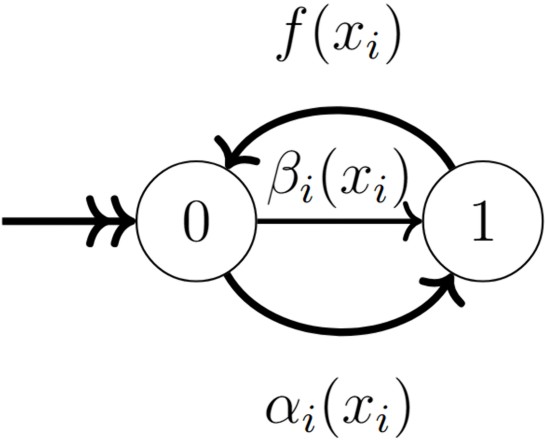

**Fig 1**. **State diagram for the binary health state of any particular player in the game.** All players start at the state 0, they can transition to the state 1 by random environmental infection, with probability $\beta_i(x_i)$ or by interacting with an infected player which will lead to an infection with probability $\alpha_i(x_i)$. Players in state 1 can transition back to state 0 with probability $f(x_i)$.

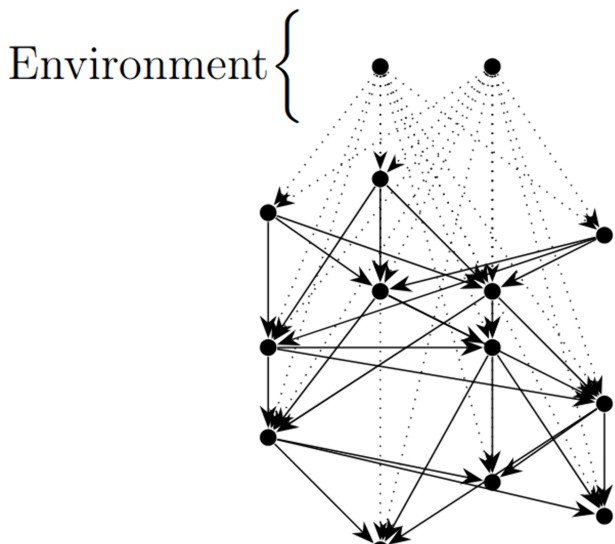

**Fig 2**. **A connected acyclic digraph representing paths of transmission through the trade network.** The environment is represented by some finite number of source terms each with possibly different different rates of infection. The weights of the digraph are represented by the weighted adjacency matrix $W$.

We assume that there are $m$ environmental sources which pose a constant threat of contagion, and so we add these as vertices with indegree 0 to our digraph. Adding such a vertex does not ruin the acyclic nature of the digraph. Call the new adjacency matrix $W \in [0, 1]^{(n+m) \times (n+m)}$ where

$$W = \begin{bmatrix} \mathbf{O} & \mathbf{O} \\ [\epsilon] & A(X, 1) \end{bmatrix}$$

where $\mathbf{O}$ is the zero matrix, and $[\epsilon]$ is the matrix describing the weight of the edges between environmental sources and players. We can add any number of environmental contagion sources and our digraph will remain acyclic. Additionally,

because it is always a source of contagion, it will always be considered infected. Let $\hat{I} \in \{0, 1\}^{n+m}$ be the concatenation of the infection state of the environment and the contagion state of the players in the game. Because the environment is a constant threat of contagion we get, if $i \leq m$,

$$P(\hat{I}_i(t+1) = 1) = P(\hat{I}_i(t) = 1) = 1.$$

At any time step, a player may become infected, may remain infected, or may recover. We are assuming that the time step in this model is very small and so the probability that more than one such event occurs in the same time step is vanishingly small. Let us first consider the likelihood of becoming infected. Vertices which had indegree 0 before the addition of the environmental vertices have a probability of becoming infected that is the sum of the probabilities of becoming infected from each environmental node.

$$P(\hat{I}_i(t+1) = 1 | \hat{I}_i(t) = 0) = \sum_{j=1}^{m} \beta_{i,j}(x_i) w_{i,j} P(\hat{I}_j(t) = 1) = \sum_{j=1}^{m} \beta_{i,j}(x_i) w_{i,j}.$$

where $\beta_{i,j}$ is a function describing how likely it is for player $i$ to take up the contagion from the environmental source $j$, given $i$'s current strategy and $w_{i,j}$ is the entry in $W$ in the $i$th row and $j$th column. This may be different for every player. For instance, in a wildlife trade network we would expect wildlife importers with low indegree to be more likely to be connected to environmental sources of contamination than players further down the network. Note that because the size of the time step is so vanishingly small we do not consider the probability of being infected by two environmental sources at the same time.

Downstream players can be randomly infected from the environment, assuming they are connected to those sources, but can also independently be infected through interacting with an infected player. The probability of player $i$ interacting with player $j$ when $j$ is infected ($a_{i,j}(X, 1)$) is given as an entry in the matrix $W$ by $w_{i+m,j+m}$. This gives us the resulting probability of infection, which is equal to the probability of becoming infected by the environment plus the probability of becoming infected by interacting with an upstream player that is currently infected.

$$P(\hat{I}_i(t+1) = 1 | \hat{I}_i(t) = 0) = \sum_{j=1}^{m} \beta_{i,j}(x_i) w_{i,j} + \sum_{j=m+1}^{i-1} \alpha(x_i) w_{i,j}(X, 1) P(\hat{I}_j(t) = 1) \qquad (2)$$

Here $\alpha_i(x_i)$ is the probability that $i$ becomes infected given that an interaction with an infected individual takes place. Notice again that we are assuming that the likelihood of being infected by multiple interactions in a single time step is negligible. Also notice that we must assume that $i$'s interaction with $j$ does not depend on the infection status of any upstream player other than $j$.

Now, let us consider the probability of recovery. For every (non-environmental) player it is the same:

$$P(\hat{I}_i(t+1) = 0 | \hat{I}_i(t) = 1) = f_i(x_i).$$

If we make the same assumption about small enough time steps so that no stakeholder can become infected and recover in the same time step (and the other way around), we see that the probability of remaining infected is the compliment of the constant probability of recovery.

$$P(\hat{I}_i(t+1) = 1 | \hat{I}_i(t) = 1) = 1 - f_i(x_i).$$

Again, note that the environmental nodes do not recover.

Notice that now we can compute for nodes $i = 1$ to $n + m$

$$P(\hat{I}_i(t+1) = 1) = P(\hat{I}_i(t+1) = 1|\hat{I}_i(t) = 0) +$$
$$P(\hat{I}_i(t+1) = 1|\hat{I}_i(t) = 1)P(\hat{I}_i(t) = 1) - \qquad (3)$$
$$P(\hat{I}_i(t+1) = 1|\hat{I}_i(t) = 0)P(\hat{I}_i(t) = 1)$$

using simple conditional probability. The probability that $i$ is infected in the next time step comes directly from the probability that they become infected given that they were not infected previously and the probability that they stay infected given that they were infected previously. With this temporal dynamic, it will be helpful to define the following

$$S := \begin{bmatrix} \mathbf{O} & \mathbf{O} \\ [\mathbf{Env(X)}] & diag(\alpha(\mathbf{X}))A(X, \mathbf{1}) \end{bmatrix} \in \mathbb{R}^{n+m \times n+m}$$

$$R := \begin{bmatrix} I_m & \mathbf{0} & \mathbf{0} & \dots & \mathbf{0} \\ \mathbf{0}^T & 1 - f(x_1) & 0 & \dots & 0 \\ \mathbf{0}^T & 0 & 1 - f(x_2) & \dots & 0 \\ \vdots & \vdots & \vdots & & \vdots \\ \mathbf{0}^T & 0 & 0 & \dots & 1 - f(x_n) \end{bmatrix} \in \mathbb{R}^{n+m \times n+m}$$

where $\mathbf{O}$ is the zero matrix, $\mathbf{0}$ is the zero vector, $[\mathbf{Env(X)}]$ is the component wise multiplication $[[\beta_{i,j}(x_i)]] * [\epsilon]$, and $diag(\alpha(X))A(X, 1)$ is the matrix product of the matrix with $\alpha_i(x_i)$ on its diagonal with the matrix $[[a_{i,j}(X, 1)]]$. Note that this product is strictly lower triangular.

$S$ can be thought of as a weighted adjacency matrix for the acyclic digraph with the environmental source, weighted by the likelihood of transmission. Let $p_t = P(I_i(t) = 1)$ and notice that the first term in (3) can be expressed as $Sp_t$, the second term can be expressed as $Rp_t$, and the final term can be expressed as $Sp_t * p_t$, where $(*)$ is component wise multiplication. This gives us the nonlinear difference equation

$$p_{t+1} = (S + R)p_t - (Sp_t) * p_t. \qquad (4)$$

Under the assumption that $\alpha(x_i) < 1$ and $\beta(x_i)$ is sufficiently small, we can work towards some nice convergence results.

Let our transformation be called $Tp := (S + R)p + (Sp) * p$. It is clear to see that, by construction, $T : \{1\} \times [0, 1]^n \to \{1\} \times [0, 1]^n$ where $\mathbf{1}$ is the vector of all 1s representing the infection state of the environmental nodes. Finding asymptotic infection probabilities across all the vertices will allow us to compute best responses directly, with the assumption that the system runs for a sufficiently long time without collapse during the transient dynamics.

Although the "next generation" operator $T$ is not linear, it is lower triangular which allows us to directly compute its fixed point. Moreover, regardless of our initial choice of $p$, applying the next generation matrix repeatedly will necessarily result in a sequence which converges to the fixed point so long as the acyclic digraph is finite. Moreover, that limit is certainly the unique fixed point of $T$ and is exactly computable through forward substitution.

**Theorem 1.** *Suppose $G$ is an acyclic weighted digraph of order $n + m$, where only $m$ vertices have indegree 0, with weighted adjacency matrix $S \in [0, 1]^{n+m \times n+m}$ with each row satisfying $\|\vec{s}\|_{L^1} \leq 1$. Further suppose $R \in [0, 1]^{n+m \times n+m}$, is a diagonal matrix with exactly the first $m$ entries equal to 1. Let the nonlinear operator $T : \{1\}^m \times [0, 1]^n \to \{1\}^m \times [0, 1]^n$ be defined as $Tp := Sp + Rp - Sp*p$ (Where * is component wise multiplication). Under these conditions, the sequence*

$(T^k p)_{k=0}^{\infty}$ converges to

$$[p^\star]_j = \begin{cases} 1 & j \le m \\ \dfrac{\sum_{i=1}^{j-1} s_{i,j} p_i^*}{1 - r_{j,j} + \sum_{i=1}^{j-1} s_{i,j} p_i^*} & j > m \end{cases} \tag{5}$$

for any $p \in \{1\}^m \times [0,1]^n$.

*Proof*: The digraph is acyclic so give the vertices a topological ordering which ensures that the vertices with indegree 0 are labeled $1,2,...,m$, the weighted adjacency matrix $S$ is strictly lower triangular, and $R$ has $r_{i,i} = 1 \iff i \le m$. Let $p \in \{1\}^m \times [0,1]^n$ so $[p]_i = 1$ for $i \le m$. Note the following two points: First, $\sum_{i=1}^{m} s_{i,j} p_i \le 1$ for all $j$ so $|r_{j,j} - \sum_{i=1}^{m} s_{i,j} p_i| < 1$ for all $j > m$. Secondly, the convergence $[T^{(k)} p]_i \to p_i^*$ is trivial for $i \le m$.

Observe that $[Tp]_{m+1} = \sum_{i=1}^{m} s_{i,m+1} + (r_{m+1,m+1} - \sum_{i=1}^{m} s_{i,m+1}) p_{m+1}$. This is because $S$ is strictly lower triangular. Moreover we see easily that applying the transformation again gives us the recurrence relation

$$[T^{(k)} p]_{m+1} = \sum_{i=1}^{m} s_{i,m+1} + (r_{m+1,m+1} - \sum_{i=1}^{m} s_{i,m+1})[T^{(k-1)} p]_{m+1}$$

Because the coefficients in this recurrence relation are constant for any choice of $m$ we will let $a_{m+1} = \sum_{i=1}^{m} s_{i,m+1}$ and $b_{m+1} = (r_{m+1,m+1} - \sum_{i=1}^{m} s_{i,m+1})$. Now if $x_k = [T^{(k)} p]_{m+1}$ we can write the recurrence relation as $x_k = a + b x_{k-1}$. Using the linearity of this difference equation, we can find a particular solution $x_k^p = \frac{a}{1-b}$ and the homogeneous solution $x_k^h = C b^k$ for some constant $C$ depending on the initial condition. We know therefore that the solution $x_k = x_k^h + x_k^p$ solves the difference equation and we get the general form

$$[T^{(k)} p]_{m+1} = C \left( r_{m+1,m+1} - \sum_{i=1}^{m} s_{i,m+1} \right)^k + \frac{\sum_{i=1}^{m} s_{i,m+1}}{1 - r_{m+1,m+1} + \sum_{i=1}^{m} s_{i,m+1}}.$$

where $C = p_{m+1} - \frac{a}{1-b}$. Thus, because $|b| = |r_{m+1,m+1} - \sum_{i=1}^{m} s_{i,m+1}| < 1$, for any $\varepsilon_1 > 0$, $\exists K_1$ such that for $k > K_1$, $|[T^{(k)} p]_{m+1} - p_{m+1}^*| < \varepsilon_1$, that is $[T^{(k)} p]_{m+1} \to p_{m+1}^*$.

Now for the inductive step we will use the elementary analysis lemma (lemma 1 in S1 Appendix). Let $l > m + 1$ and notice that

$$[T^{(k+1)} p]_l = \sum_{i=1}^{l-1} s_{i,l}[T^{(k)} p]_i + \left( r_{l,l} - \sum_{i=1}^{l-1} s_{i,l}[T^{(k)} p]_i \right)[T^{(k)} p]_l$$

Let $y_k := [T^{(k+1)} p]_l$ and let $[T^{(k)} p]_i$ for $i = 1, ..., l-1$ together form the vector $\vec{x}_k$. Further let, $a(\vec{x}) = \sum_{i=1}^{l-1} s_{i,l} \vec{x}_i$ and $b(\vec{x}) = r_{l,l} - \sum_{i=1}^{l-1} s_{i,l} \vec{x}_i$. $a, b \in C^0(\mathbb{R}^{l-1}, \mathbb{R})$. Thus, if we rewrite the above equation as

$$y_k = a(\vec{x}_{k-1}) + b(\vec{x}_{k-1}) y_{k-1}$$

and note that $\lim \vec{x}_k = \vec{x} := [p_i^*]_{i=1}^{l-1}$, the lemma says that $y_k \to \frac{a(x)}{1-b(x)}$ as $k \to \infty$. Translating back, this means that

$$\lim_{k \to \infty} [T^{(k)} p]_l = \frac{\sum_{i=1}^{l-1} s_{i,l} p_i^*}{1 - r_{l,l} + \sum_{i=1}^{l-1} s_{i,l} p_i^*} = p_l^*$$

This forms the basis for our inductive argument. Because $[T^{(k)}p]_i = 1$ for all $k$ when $i \leq m$ and because $[T^{(k)}p]_{m+1} \to p^*_{m+1}$, the inductive step tells us that $[T^{(k)}p]_{m+2} \to p^*_{m+2}$. We may repeat this a *finite* number of times to see that $[T^{(k)}p]_i \to p^*_i$ for all $i = 1, ..., n + m$ which we write as $\lim_{k \to \infty} T^{(k)}p = p^*$

This works regardless of our initial choice for $p$ and so we have shown the desired result. □

This result allows us to compute exact expected values for $\pi(X, l)$ and easily approximate (or, with great effort, compute exactly) how payoffs change with respect to the components of the strategy. In this way, we can find best responses. This provides us a direct mapping from the set of global parameters and vertex specific parameters to payoffs $w : \mathbb{R}^{n \cdot l} \to \mathbb{R}^n$ where $l$ is the dimension of the strategy vector $x_i$.

$$w(X) = g(X, P^*(X)) \tag{6}$$

We can be more exact about what $g$ looks like. On expectation the probability that $i$ will interact with $j$ given that $j$'s infection status is $p^*_j$ is simply $p^\star_j a_{i,j}(X, 1) + (1 - p^*_j)a_{i,j}(X, 0)$. Therefore we can make this substitution into our original payoff function (1) to get

$$\begin{aligned}
w_i(x_i, X_{-i}) = {} & p^*_i d_i(x_i, 1) + (1 - p^*_i)d_i(x_i, 0) \\
& + \sum_{j=1}^{i-1} c_{i,j}(x_i, x_j)(p^*_j a_{i,j}(X, 1) + (1 - p^*_j)a_{i,j}(X, 0)) \\
& + \sum_{j=i+1}^{n} b_{i,j}(x_i, x_j)(p^*_i a_{j,i}(X, 1) + (1 - p^*_i)a_{j,i}(X, 0))
\end{aligned} \tag{7}$$

This means that we can carry out sensitivity analyses on the global parameters and the graphical structure and easily measure things like long time infection probabilities across the whole network.

## 3 Equilibrium results

Equipped with the result from Theorem 1, we can turn our attention to the analysis of this system. In order to find Nash equilibria, we must discuss best responses. In particular, we seek a function $\psi_i : \mathbb{R}^{s \cdot (n-1)} \to \mathbb{R}^s$ which, given a strategy profile $X_{-i}$ returns player $i$'s best response. If such a function exists, we can go further to find a function $\Psi(X) = [\psi_i(X_{-i})]_{i=1}^n$, which takes a strategy profile $X$ and returns a new strategy profile in which every player has updated their strategy to take on a best response to the previous strategy profile. This is the Myopic Best Response (MBR) process. Surely fixed points of $\Psi$ are Nash equilibria of the game (This is direct from the definition of Nash equilibrium).

The particular form of $\Psi$ depends a great deal on the forms of the functions mentioned in Sects 2.1 and 2.2. For this reason, we do not attempt to find general results about $\Psi$ but we do present an argument for the existence of such a $\Psi$ under certain Hypotheses.

### 3.1 Existence of a best response function

We would like to discuss the existence of $x_i$ which maximizes payoff $w_i(x_i, X_{-i})$. In the case that $X \in \Omega \subset \mathbb{R}^{x \times n}$ where $\Omega$ is compact, then continuity of $w$ is sufficient for the existence of at least one maximizer of $w_i(x_i, X_{-i})$ through a "continuous on compact" argument.

In order to show the existence of a maximizer in the case where strategies are unbounded, we need some restrictions on the behavior of the functions $b$, $c$, and $d$, the functions which describe the upstream downstream and intrinsic payoff for a player. We will do this by considering the typical economic assumptions of *increasing marginal costs* and *decreasing marginal benefit*. Most simply, for our function $d$ we will require that $d_x(x_i, 1)$ and $d_x(x_i, 0)$ are both eventually decreasing in every component of $x_i$. Recall $d$ is the function which describes the component of payoff which is independent of other

players. If we consider $x_i$ as different components of investment into different elements relating the game, then the laws of increasing marginal cost and decreasing marginal benefit will impose the assumption that eventually, increasing any investment will decrease payoff.

Recall that $c_{i,j}(x_i, x_j)$ governs the payoff to player $i$ from upstream interaction (i.e., purchasing). We can assume that $c_{i,j}$ is everywhere negative. Likewise recall that $b_{i,j}$ governs the payoff to player $i$ from downstream interaction (i.e., selling). We assume that $b_{i,j}$ is everywhere positive. A simple way we can ensure the existence of a best response is by assuming that both $b_{i,j}(x_i, x_j)$ and $c_{i,j}(x_i, x_j)$ are bounded in the first argument for all $i, j$. The assumption can be justified through the finiteness of resources available to stakeholders. Surely each player has a limit to how much they can buy from players upstream and how much they can sell to players downstream. These two assumptions together are the basis for Hypotheses 2 and 3 in the following theorem, forming a loose set of reasonable sufficient conditions for the existence of a best response function.

Under these assumptions, with a strategy space equipped with tie-breaking order, there then exists a function $\psi_i : \mathbb{R}^{s \times (n-1)} \to \mathbb{R}^s$ such that $\psi_i(X_{-i})$ is $i$'s best response to $X_{-i}$. Let $\Psi : \mathbb{R}^{s \times n} \to \mathbb{R}^{s \times n}$ be the concatenation of each of these functions so that $\Psi(X) = [\psi_i(X_{-i})]_{i=1}^n$. This is written formally in Theorem 2. It is immediate from the definition of Nash equilibrium that a fixed point of $\Psi$ is a Nash Equilibrium of the game. For use in modeling, finding fixed points of $\Psi$ is crucial and can be done numerically.

**Theorem 2.** *Let $\Omega \subseteq \mathbb{R}^{s \times n}$ be a strategy space for the game defined by the payoff Eq* (1)*, equipped with a tie breaking ordering, $\prec_\Omega$. Under the following hypotheses*

H1) *$a_{i,j}, b_{i,j}, c_{i,j}, \beta_{i,j}, d_i, f_i,$ and $\alpha_i$, as defined above, are all continuous in $x_i$ for all $i, j = 1, 2, ..., n$*
H2) *There exits an $R$ such that $|x_i| > R \implies d_i(x_i, I_i) \leq M_0 - \epsilon |x_i|$ for some $M$ and some $\epsilon$.*
H3) *$|c_{i,j}|$ and $|b_{i,j}|$ are bounded by $M$*

*the function $\Psi : \Omega \to \Omega$ which satisfies the condition that $\Psi(X)$ is the concatenation of every players' best response to $X$ is well defined. Moreover, if $\Omega$ is compact, then (H2) and (H3) are unnecessary.*

*Proof*: let $\Omega = \Omega_i \times \Omega_{-i}$ where for any player $i$, $x_i \in \Omega_i$ and $X_{-i} \in \Omega_{-i}$. Payoff for each player $i$ is determined by $w_i(x_i, X_{-i})$ as in (7) For any particular $X_{-i}$ we seek to show that that the maximum of $w_i(x_i, X_{-i})$ over $x_i \in \Omega_i$ is attained.

When $\Omega \subset \mathbb{R}^{s \times n}$ is compact then we need only show that $w_i$ is continuous. From (H1) it is obvious that $\pi_i(x_i, X_{-i}, 1)$ is the sum of continuous functions and so it itself continuous. Moreover, $p_i^\star(x_i, X_{-i})$ for any player $i$ is given as (5). Again, it is clear that from (H1) that $s_{i,j}$ is continuous for all $i, j = m + 1, m + 2, ..., m + n$ (corresponding to all the players). If $f$ is continuous then so is $r_{i,i}$ and, as noted in Theorem 1 the denominator of (5) is always strictly positive. Therefore the quotient of these two continuous functions is continuous and thus so is $p_i^\star$. With the continuity having been determined, it is direct from the compactness of $\Omega$ that there is an $x_i^\star \in \Omega_i$ such that $w_i(x_i^\star, X_{-i}) = \max_{\Omega_i} w_i(\cdot, x_{-i})$. The fact that the argmax of $w_i(\cdot, X_{-i})$ is nonempty means that the we certainly have a best response by way of the tie-breaking order $\prec_\Omega$. Clearly a tie-breaking order is certain to exist so long as the strategy space has finite dimension.

In the case that $\Omega$ is not compact, we put some decay estimates on $w_i$ so that we know the maximum is attained. Notice that $w_i$ is a convex combination of terms from $\pi_i(x_i, X_{-i}, 1)$ and $\pi_i(x_i, X_{-i}, 0)$ and so, obviously,

$$\min_{I \in \{0,1\}^n} \pi_i(x_i, X_{-i}, I) \leq w_i(x_i, X_{-i}) \leq \max_{I \in \{0,1\}^n} \pi_i(x_i, X_{-i}, I).$$

By (H3) it is easy to see that $\pi_i(x_i, X_{-i}, I) \le d_i(x_i, I_i) + Mn$ for any $I \in \{0,1\}^n$. Let $m := \min_{I \in \{0,1\}^n} \pi_i(\mathbf{0}, X_{-i}, I)$. Let $R_1 = \max\{R, \frac{-m + M_0 + Mn}{\epsilon}\}$. When $|x_i| > R_1$, then $d_i(x_i, I) < m - Mn$ and thus $\pi(x_i, X_{-i}, I) < m$, for any $I$. This implies that

$$\max_{x_i \in \Omega \setminus B_{R_0}(0)} \pi_i(x_i, X_{-i}, I) < m \le \max_{x_i \in \overline{B_{R_0}(0)}} \pi_i(x_i, X_{-i}, I)$$

for any $I$. This means that outside of $B_{R_0}(0)$, $w_i(x_i, X_{-i}) < m$ and, because we now have that $m \le \sup_\Omega w_i(x_i, X_{-i})$, we can say that $\sup_\Omega w_i(x_i, X_{-i}) = \max_{\overline{B_{R_0}(0)}} w_i(x_i, X_{-i})$, which surely exists by the same continuity on compact argument. Now that we know that the maximum of $w(x_i, X_{-i})$ is attained on $\Omega$, then surely we know that, given the tie breaking order $\prec_{\Omega_i}$, every $X_{-i}$ corresponds to one best response $x_i$.

Thus we have shown that for any $X_{-i}$, whether $\Omega$ is compact or not, there exists a best response. Let $\psi_i : \mathbb{R}^{s \times (n-1)} \to \mathbb{R}^s$ be the function which maps $X_{-i}$ to its best response. This is well defined (although we have no hope of describing its behavior or regularity). Easily concatenate these functions for all $i$ to get $\Psi : \mathbb{R}^{s \times n} \to \mathbb{R}^{s \times n}$ where $\Psi(X) = [\psi_i(X_{-i})]_{i=1}^n$. This function is well defined, and maps a strategy profile $X$ to the strategy profile that where each player is playing a best response to $X$. □

Because we have only shown that the best response function is well defined, but have done nothing to describe its behavior or regularity, we can do very little to determine the existence of a fixed point much less the stability of such a fixed point. However, it does mean that the process of myopic best response is well defined and that there is, perhaps, a process by which fixed points may be found numerically. In $\Omega$ we may solve the problem $X - \Psi(X) = 0$. Without regularity of $\Psi$, or indeed the a priori knowledge that a solution exists, we are not guaranteed convergence of the myopic best response process, but we can use it to examine the evolution of strategies in time.

## 4 A numerical example through myopic best response

Here we present a toy example to demonstrate that this method of considering trade network games can be used to reveal surprising results about how the structure of a trade network may impact the strategic equilibria. Consider a game with $N$ players organized in an acyclic digraph with a strictly lower triangular adjacency matrix $W$ where each row sums to 1. This is a trade network where each edge describes an "upstream" transaction (i.e., a purchase). As before suppose each player has a one dimensional strategy $x_i \in [0, 1]$ and an infection state $I_i \in \{0, 1\}$. In this toy example, "strategy" is mildly modeled after "investment into health and safety measures" but is far too simple to capture that fully.

Suppose that the likelihood of transaction $a_{ij}(X, I_j) = w_{i,j}(1 - x_i I_j)$ so as player $i$'s investment increases, their likelihood of interacting with an infected player decreases. Recall that total payoff can be separated into three components: The benefit to player $i$ from interacting upstream (purchasing), the benefit to player $i$ from interacting downstream (selling), and the intrinsic benefit to player $i$. Consider the game where each of these components for player $i$ interacting with a player $j$ are defined as follows

$$\text{downstream:} \quad b(x_i, x_j) = (1 - x_i)$$
$$\text{upstream:} \quad c(x_i, x_j) = (x_j - 1)$$
$$\text{intrinsic:} \quad d(x_i, I_i) = x_i(1 - x_i)(1 - I_i)$$

Notice that the upstream and downstream components satisfy the relation $c(x_i, x_j) = -b(x_j, x_i)$; the costs of the interaction to the recipient are the same as the benefits to the source. When we plug these functional forms into the payoff function we can interpret each of the terms as follows. For a downstream interaction, (the final sum in Eq (8)), if player $i$ can assume player $j$ will necessarily interact with them, $i$ can maximize their payoff from that interaction by minimizing their investment in health and safety. However, when that investment is lowered, probability of infection goes up which decreases $j$'s probability of purchasing from $i$ ($w_{j,i}(i - x_j I_i)$) whenever $x_j > 0$. For an upstream interaction (The middle term

in (8)), if a player $i$ invests a lot in health and safety, they will be very unlikely to purchase from player $j$ if $l_j = 1$ but if such a purchase occurs, the benefit to player $i$ is maximized when player $j$ invests a lot in health and safety. The intrinsic benefit (the first term in (8)) for a player is minimized in this toy example when the player does not invest in health and safety (reflecting personal beliefs, perhaps) or when the player invest at the highest level in health and safety (reflecting costs of such investment). It is maximized at the intermediate level of investment which minimized the probability of infection. With each of these components, we can compute the total payoff as

$$\pi(x_i, x_{-i}, l) = d_i x_i (1 - x_i)(1 - l_i) + \sum_{j=1}^{i-1} (x_j - 1) w_{i,j} (1 - x_i l_j)$$

$$+ \sum_{j=i+1}^{N} (1 - x_i) w_{j,i} (1 - x_j l_i)$$

(8)

where $d_i$ is the intrinsic benefit weight. One of the major issues with game theoretic models with intrinsic benefit is that determining how important the intrinsic benefit is relative to the extrinsic factors can be very difficult, especially when the intrinsic benefit is measured in a qualitative measure like "enjoyment" while the extrinsic factors are attached to a currency. From a modeling perspective, this is an issue of unit analysis. If $d_i(x_i, l_i)$ from Eq (1) has units of "enjoyment" while $c_{i,j}(x_i, x_j)$ and $b_{i,j}(x_i, x_j)$ have units of currency, we must include a conversion factor which relates the currency to the enjoyment. To account for this, we allow for a change in weight of the intrinsic benefit. if $d_i = 0$ then player $i$ does not consider the benefit of having any stock, if $d_i$ is very large, the benefit of having a healthy stock outweighs any extrinsic factors. We also make the decision that $\alpha_{i,j}(x_i) = \beta_{i,j}(x_i) = (1 - x_i)$ and that the probability of recovery is exactly $f(x_i) = x_i$ With these assumptions, we can write down our process for computing $p^*$ as described in Theorem 1.

$$p_i^* = \begin{cases} 1 & i = 1 \\ \dfrac{\epsilon(i - x_i) + \sum_{j=1}^{i-1} w_{i,j}(1 - x_j)^2 p_j^*}{x_i + \epsilon(i - x_i) + \sum_{j=1}^{i-1} w_{i,j}(1 - x_j)^2 p_j^*} & i > 1 \end{cases}$$

From this, we can write the long-run payoff function $w$ as

$$w(x_i, x_{-i}) = d_i x_i (x_i - 1)(1 - p_i^*)$$

$$+ \sum_{j=1}^{i-1} (x_j - 1) w_{i,j} (1 - x_i p_j^*)$$

$$+ \sum_{j=i+1}^{n} (1 - x_j) w_{j,i} (1 - x_j p_i^*)$$

Notice that, because of the linearity of this system, we can write $w_i(x_i, x_{-i}) = \pi_i(x_i, x_{-i}, p_i^*)$. This is not generally true when the functional forms are non-linear. Because we now have a map $w : [0, 1]^n \to \mathbb{R}^n$ which takes the strategy profile $X \mapsto w(X)$ a vector of fitnesses, we can use Theorem 2 to show that the map $\Psi : [0, 1]^n \to [0, 1]^n$ which takes a strategy profile $X$ and returns a best response, $\Psi(X)$, is well defined given a certain tie-breaking order, $<_\Omega$. Although we cannot investigate this function analytically in this example, we can investigate it numerically to approximate fixed points of the map $\Psi$. We repeatedly optimize $w_i(x_i, X_{-i})$ with respect to the first variable for each $i$, starting from many initial strategy profiles because there is no guarantee that $\Psi$ has a unique fixed point. This process is exactly the myopic best response process from evolutionary game theory and will terminate in a Nash equilibrium (if it terminates).

## 4.1 Proof of concept on symmetric and asymmetric networks

As a sanity check we show that network structure has measurable effects on the strategic equilibria of the game. Consider two networks which have the same sets of players but slightly different patterns of interaction.

We will use the example of two nearly identical three layer trade networks that differ only in the edges between levels two and three (Fig 3). In the symmetric case there are two producers, two distributors (s1 and s2) which each buy from both producers evenly, and four consumers. One consumer buys only from s1, one buys only from s2 and the other two buy from both s1 and s2 evenly. In the asymmetric case, again there are two producers, two distributors (a1 and a2) which buy from the producers evenly and four consumers. However, in this network three consumers buy only from a1 and the fourth buys evenly from a1 and a2. In the asymmetric network, a2 has only one potential customer.

The results, summarized in Table 1 for the distributors, show that, as expected, the structure of the network has a reasonable impact on the results of the game. It is a helpful sanity check to confirm that symmetric conditions result in symmetric solutions. It is also important to note that the numeric results seem to support the monostability of the system as the same equilibrium solutions are found in both the asymmetric and symmetric cases regardless of the initial data (plus or minus some numerical error which was observed as high as $5 \times 10^{-4}$, perhaps because of the rather inelegant method of optimization used here).

The main result is that the symmetric system resulted in symmetric results and that in the asymmetric system, a1, the sole distributor to the majority of the consumers, had a lower strategy at equilibrium than a2. It was also observed that the long term probability of infection for a1 was higher than that of a2. This is discussed further in Sect 5.

In global terms we may also want to consider the total probability of infection. The naïve version considers the probability that none of the stakeholders are infected, which can be given an easy upper bound. We take the "Naïve risk" to be $1 - \prod_{i=1}^{n}(1 - p_i^*)$, which is an upper bound on the probability that at least one individual is infected. One can see that it is an upper bound in the following way. Suppose that the probability that player $i$ is not infected is given as $P(A_i)$ and thus

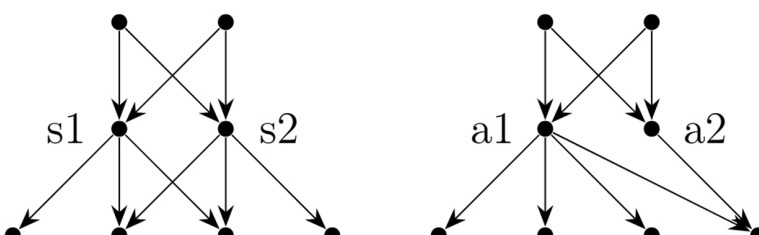

**Fig 3**. **Two trade networks with 8 players.** The arrows point in the direction that goods are passed through the system (which is different from the way that the adjacency matrix $W$ encodes the information). On the **left** is the symmetric case and on the **right** is the asymmetric case. The two nodes of interest in the symmetric case are labeled $s1$ and $s2$ and in the asymmetric case those same players are labeled $a1$ and $a2$.

**Table 1**. **The results of the equilibrium search through myopic best response on the symmetric and asymmetric network reported for the focal individuals s1, s2, a1, and a2.** In the symmetric case, the equilibrium strategies are equal while in the asymmetric case, the distributor with the greater "market share" has a lower strategy (which again can be considered low investment in health and safety measures for the purposes of application) than the one with the smaller market share. The probabilities of infection at equilibrium are also reported for each player. In the symmetric game both players have the same probability of infection, in the asymmetric game the distributor with the greater market share has a higher probability of infection. Risk measurements for each network are reported. Regardless of which measure of risk is chosen, the asymmetric network poses a greater risk.

|  | Symmetric | | Asymmetric | |
| --- | --- | --- | --- | --- |
|  | s1 | s2 | a1 | a2 |
| Equilibrium Strategy | 0.4377 | 0.4377 | 0.3924 | 0.5195 |
| Infection Probability | 0.1944 | 0.1944 | 0.2314 | 0.1391 |
| Naïve Risk | 0.7276 | | 0.7298 | |
| Weighted Risk | 0.4614 | | 0.5365 | |

the probability that no player is infected can be given as $P(\bigcap_{i=1}^{n} A_i)$. We write this as

$$P\left(\bigcap_{i=1}^{n} A_i\right) = P(A_1)P(A_2|A_1)P(A_3|A_2 \cap A_1)P(A_n|A_{n-1} \cap A_{n-2} \cap \cdots \cap A_1) \qquad (9)$$

We will note that the probability that player $i$ is infected can only increase when a player upstream of $i$ is infected. Thus we can say $P(A_2^c|A_1^c) \geq P(A_2^c|A_1)$. This implies that $P(A_2|A_1) \geq P(A_2|A_1^c)$ and so we get

$$P(A_2) = P(A_1)P(A_2|A_1) + P(A_1^c)P(A_2|A_1^c)$$
$$\leq P(A_1)P(A_2|A_1) + (1 - P(A_1))P(A_2|A_1)$$
$$\leq P(A_2|A_1)$$

We can repeat this for every element in the product (9) and get

$$P\left(\bigcap_{i=1}^{n} A_i\right) \geq \prod_{i=1}^{n} P(A_i) \implies 1 - P\left(\bigcap_{i=1}^{n} A_i\right) \leq 1 - \prod_{i=1}^{n}(1 - P(A_i^c)).$$

Therefore we can use our easy to compute measure of risk as an upper bound for probability that at least one individual is infected.

In this example the symmetric case has a naïve risk of 0.7276 and the asymmetric case has a naïve total probability of 0.7298. However, this approach does not account for the fact that infections at the top of the trade network pose a greater threat to the network as a whole. Because the naïve metric lacks exactness and is unable to account for network position, we next develop a more robust, weighted metric of risk.

Recall that $A(X, I_j)$ is the weighted adjacency matrix describing the probability of transacting upstream given a strategy profile $X$ and an infection profile $I$. Suppose that individual $i$ is infected. The expected number of additional infections $i$ may cause after a single transaction are given by the sum of the $i^{th}$ column of the matrix $S$ where $s_{i,j} = \alpha(x_i)a_{i,j} = w_{i,j}(1 - x_i)^2$. Likewise the expected number of additional infections $i$ may cause *indirectly* from a sequence of two transactions is given by the sum of the $i^{th}$ column of $S^2$. We can repeat this process for increasing path lengths until eventually $S^k = O$ (which is certain to happen because $S$ is strictly lower triangular and thus nilpotent of degree $\leq n$.) By adding each of these expected number of additional infections we get an estimation of $\mathcal{R}_0^i$ for each player $i$.

$$\mathcal{R}_0^i = \sum_{k=1}^{n} \mathbf{1}^T S^n \hat{e}_i$$

This quantity represents the total number of expected new infections $i$ may cause should $i$ itself become infected. This gives us our weighted risk measurement which we call $\mathcal{R}_g$ which we use as a measure of spillover risk from the network. Although more sophisticated measures of spillover risk exist, for our purposes this weighted risk measurement gives an appropriate estimation of the number of new infections appearing in the long run equilibrium state. If we assume each infection behaves as an independent opportunity to yield a spillover event, then this equilibrium state will be approximately proportional to the number of spillover cases.

$$\mathcal{R}_g = \sum_{i=1}^{n} \mathcal{R}_0^i p_i^*.$$

Applying this measurement to our toy example we see that the symmetric case has a weighted risk of 0.4614 and the asymmetric case has a weighted risk of 0.5365. Assured that our model works and can be reasonably assessed, we can proceed to discuss other questions about the toy model.

## 4.2 Monopoly effects in the toy model

To consider how market share plays a roll in infection risk for a wildlife trade network, we consider three networks with 15 players (Fig 4). In case there are three distributors who each buy evenly from the two producers. The difference is in how the 10 consumers buy from the three distributors. In the competitive case every consumer buys evenly from each of each distributor (c1, c2, c3). This means that each of the distributor has exactly one third of the market share. In the mild monopoly case two consumer buy evenly from distributors mm1 and mm2, two of them buy evenly from mm2 and mm3, and the remaining consumers buy exclusively from mm2. This means that mm1 and mm3 each have 10% of the market share and mm2 has 80% of the market share. Lastly, in the total monopoly case all ten consumers buy only from tm2, none from tm1 and tm3. Clearly this means that tm2 has 100% of the market share and tm1 and tm3 act as terminal consumers in the game.

As the monopolistic quality of the trade network increases we see that the naïve risk decreases while the weighted risk increases. This can be explained by the fact that in a monopoly, the majority of the pathways of infection must pass through a single individual, the monopolist. This may mean that the distributors with less market share can choose strategies which result in a lower probability of infection so the naïve risk decreases. However, the weighted risk captures the concentration of infective pathways through the monopolist and so weights their probability of infection higher. There results are summarized in Table 2.

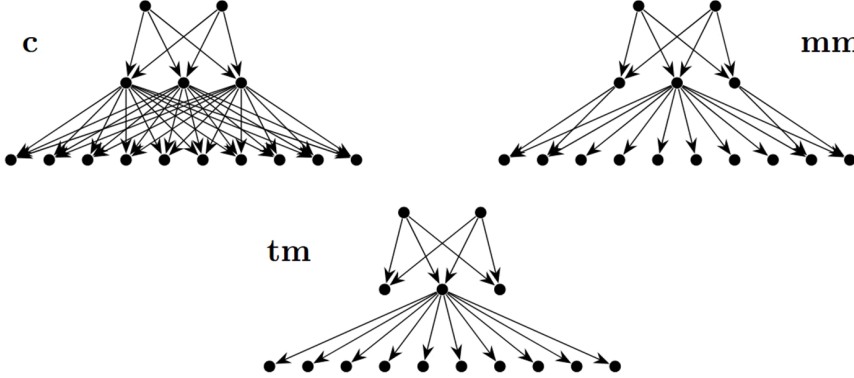

**Fig 4. Three trade networks with 15 players.** The arrows point in the direction that goods are passed through the system (which is different from the way that the adjacency matrix $W$ encodes the information). On the **top left** is the competitive case, on the **top right** is the mild monopoly case, and on the **bottom** is the total monopoly case. We will be interested in the "distributors" (middle row in each network) which are called "c1,c2, and c3" from left to right in the competitive case, "mm1, mm2, and mm3" from left to right in the mild monopoly case, and "tm1, tm2, and tm3" from left to right in the total monopoly case.

**Table 2. A comparison of the strategies, infection probabilities, and risk measurements from the networks shown in Fig 4.**

|  | Competative | | | Mild Monopoly | | | Total Monopoly | | |
|---|---|---|---|---|---|---|---|---|---|
|  | c1 | c2 | c3 | mm1 | mm2 | mm3 | tm1 | tm2 | tm3 |
| Equilibrium Strategy | 0.3451 | 0.3451 | 0.3451 | 0.4334 | 0.3082 | 0.4334 | 0.5506 | 0.3027 | 0.5506 |
| Infection Probability | 0.3734 | 0.3734 | 0.3734 | 0.2628 | 0.4220 | 0.2628 | 0.1549 | 0.4031 | 0.1549 |
| Naïve Risk | 0.9830 | | | 0.9771 | | | 0.9691 | | |
| Weighted Risk | 3.2393 | | | 3.2525 | | | 3.3292 | | |

In this game, it is also the case that having more consumers allows a distributor to take on a lower strategy which sometimes results in a higher probability of infection. For example compare c2, mm2, and tm2, as the market share increases the strategy they choose (0.3451, 0.3082, 0.3072 respectively). Interestingly this does not translate directly to a higher probability of infection. In the case of the total monopoly, tm2 is actually less likely to become infected than mm2 because of the upstream effects of the monopoly on the producers.

In a real trade network, not all stakeholders can be neatly sorted into different categories like producer, distributor, or consumer. Many stakeholders take on more than one role. This makes the results slightly harder to discuss because we can not take neat steps up and down the trade network as we can here. However, the model presented here in no way depends on the separation of stakeholders in to these categories. The only restrictive assumption is that the flow of goods is acyclic in the network.

### 4.3 Upstream cascades from parameter adjustments

Another important note about the model is that adjustment to parameters downstream can have real effects on the equilibrium strategies upstream. To demonstrate this we vary the weight on the intrinsic benefit term for the consumers, $d_i$, in the networks shown in Fig 3. As we change this weight from 0 (in the case where no consumer receives any payoff other than the cost of transacting with distributors) to 1.5 (in the case where the intrinsic benefit absolutely outweighs the cost of transacting with distributors) we can see qualitative shifts in the behavior of distributors and even producers.

Some aspects of the behavior shown in Fig 5 are hard to describe, especially because the toy model is not complex enough to capture real trade network-like behavior, but the main point is clear. As intrinsic benefit is weighted less for consumers, their equilibrium strategy decreases. This is to be expected because, in the extreme limiting case without an intrinsic benefit, there is no cost of infection for consumers. The initially surprising result, which seems natural once considered further, is the existence of tipping points, the very rapid changes in equilibrium strategy taken on by the distributors and the producers. When intrinsic benefit weight shrinks below a particular threshold, distributors are no longer incentivized to invest in health and safety measures and so their equilibrium strategy drops to zero. Around the same point, the equilibrium strategy of the producers also decreases. This change for the produces is more pronounced in the symmetric case than in the asymmetric case because in the asymmetric case, the distributor which sells only to a single individual (a2) must maintain a higher equilibrium strategy and therefore provides an incentive for the producers to maintain a higher equilibrium strategy. We do not, at this moment, have an explanation for the sudden uptick in producer strategy when intrinsic benefit approaches 0.

The coupled changes in equilibrium strategies with changing weights of the intrinsic benefit demonstrates the model's usefulness in considering interventions for disease. Throughout a trade network, different stakeholders have to play by different rules so controls at one layer of a network may be more feasible than others. By demonstrating that the entire network is sensitive to changes to parameters at a single level, we demonstrate the usefulness of this model for that purpose. This idea is discussed further in Sect 5.

### 4.4 Defectors in the toy model

Finally, we want to see the role that Defectors have in this model. In this case, we do not mean that each stakeholder has the choice to cooperate or defect, rather we mean that a single stakeholder may choose to take on a particular strategy, regardless of the payoff, because of some behavior or belief not captured by the model. For instance, choosing not to pasteurize milk because of a scientifically inaccurate political belief.

In the first case, we take the symmetric network in Fig 3 and consider a consumer, who interacts with both distributors evenly. We measure average equilibrium behavior in the case that the consumer plays rationally, they defect to the strategy $x = 0$ or they defect to $x = 1$. The results are summarized in Table 3. In Tables 4 and 5 we present the results from the cases where a distributor or producer, respectively, defects.

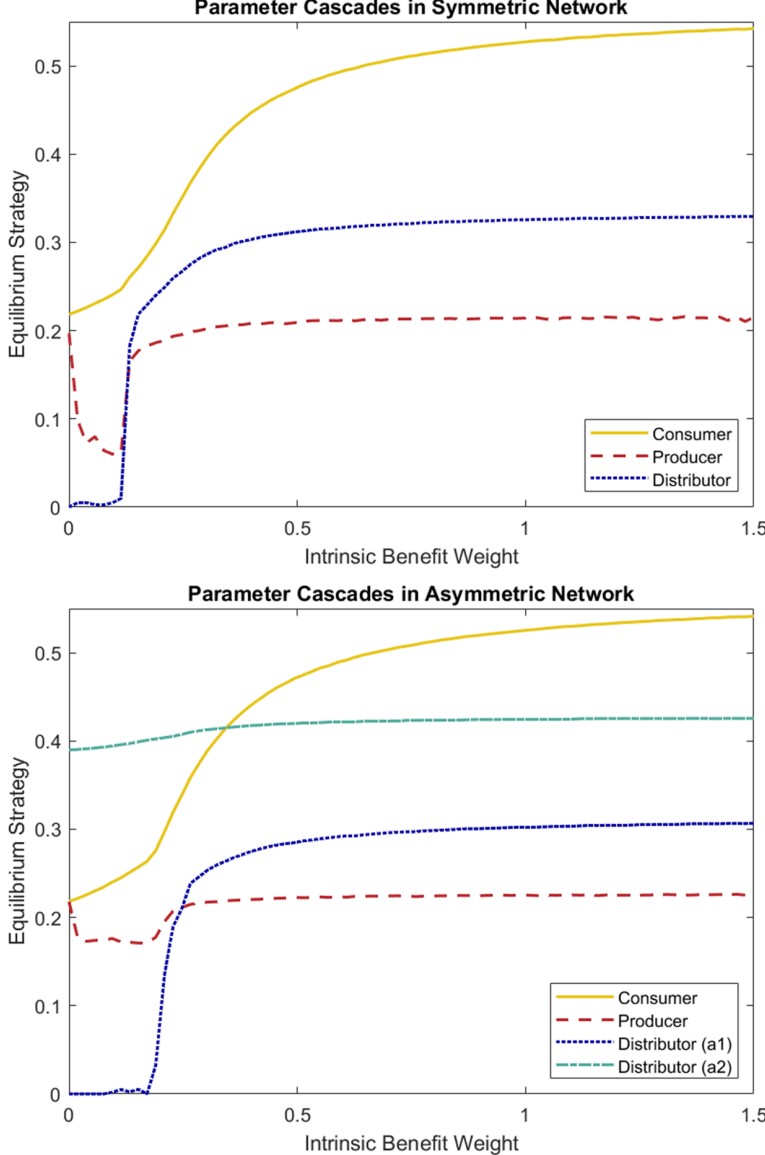

**Fig 5**. **Changing the intrinsic benefit weight uniformly for all consumers from 0 to 1.5 shows a qualitative change in strategy for both symmetric (top) and asymmetric (bottom) examples.** The consumer equilibrium strategy changes continuously with weight, the distributor strategy changes sharply at a tipping point between 0.2 and 0.3 and the producer strategy changes abruptly at the same tipping point although this change appears to be less pronounced in the asymmetric network because of the insulating effects of the asymmetry discussed previously.

The main result that we see from each situation is that a single defector can do very little to change the equilibrium strategy of players up- or downstream but they can have a great effect on the downstream infection probability. For this reason, we see that single defectors that are not terminal consumers can change the weighted risk of a network to an extreme degree. The naïve risk is nonsensical in this comparison because when a defector defects to the strategy $x = 0$ they will surely become infected and so the naïve risk will always be 1 in this case.

**Table 3**. **A comparison of the strategies, infection probabilities, and risk measurements resulting from a defecting consumer in the symmetric trade network in Fig 3.** Equilibrium Strategy and Infection Probability are listed for a rational consumer (c), a distributor (d) and a producer (p).

| | Rational | | | Defect to 0 | | | Defect to 1 | | |
|---|---|---|---|---|---|---|---|---|---|
| | c | d | p | c | d | p | c | d | p |
| Equilibrium Strategy | 0.5666 | 0.4374 | 0.3505 | 0.5668 | 0.4227 | 0.3556 | 0.5664 | 0.4444 | 0.3470 |
| Infection Probability | 0.1236 | 0.1947 | 0.1564 | 0.1261 | 0.2052 | 0.1540 | 0.1228 | 0.1907 | 0.1595 |
| Naïve Risk | 0.7277 | | | 1 | | | 0.6879 | | |
| Weighted Risk | 0.4618 | | | 0.4867 | | | 0.4555 | | |

**Table 4**. **A comparison of the strategies, infection probabilities, and risk measurements resulting from a defecting distributor in the symmetric trade network in Fig 3.** Equilibrium Strategy and Infection Probability are listed for a consumer (c), a rational distributor (d) and a producer (p).

| | Rational | | | Defect to 0 | | | Defect to 1 | | |
|---|---|---|---|---|---|---|---|---|---|
| | c | d | p | c | d | p | c | d | p |
| Equilibrium Strategy | 0.5666 | 0.4374 | 0.3505 | 0.5650 | 0.4351 | 0.3280 | 0.5695 | 0.4244 | 0.3865 |
| Infection Probability | 0.1236 | 0.1947 | 0.1564 | 0.2178 | 0.2031 | 0.1470 | 0.0970 | 0.1958 | 0.1371 |
| Naïve Risk | 0.7277 | | | 1 | | | 0.6018 | | |
| Weighted Risk | 0.4622 | | | 2.4849 | | | 0.2671 | | |

**Table 5**. **A comparison of the strategies, infection probabilities, and risk measurements resulting from a defecting producer in the symmetric trade network in Fig 3.** Equilibrium Strategy and Infection Probability are listed for a consumer (c), a distributor (d) and a rational producer (p).

| | Rational | | | Defect to 0 | | | Defect to 1 | | |
|---|---|---|---|---|---|---|---|---|---|
| | c | d | p | c | d | p | c | d | p |
| Equilibrium Strategy | 0.5666 | 0.4374 | 0.3505 | 0.5666 | 0.4400 | 0.3521 | 0.5671 | 0.4274 | 0.3579 |
| Infection Probability | 0.1236 | 0.1947 | 0.1564 | 0.1619 | 0.3503 | 0.14559 | 0.1147 | 0.1606 | 0.1525 |
| Naïve Risk | 0.7277 | | | 1 | | | 0.6338 | | |
| Weighted Risk | 0.4622 | | | 2.1790 | | | 0.314 | | |

## 5 Discussion and applications

Many individual aspects of strategy and payoff in trade networks with contagion have been studied independently. Building on the diversity of studies that have considered some subsets of factors relating to infection risks in animal trade networks [3,5,11] and the economic factors that motivate human management/trade decisions about them [3,8,15,17], we have here proposed a model that allows for consideration of the confluence of all of these factors together. In this work, we have addressed a gap in understanding the combined effects of each of these individual factors. While earlier related works have focused on estimating statistical likelihoods for spillover risks based on analysis and projection of observational data (e.g., [14]), or on characterizing specific trade network topologies (e.g., [4]), or on calculating the economic incentives of the overall system rather than of individuals in a network (e.g., [15]), we instead employ game theory on networks methods. This allows us to consider the structure of the trade network as an independent variable and observe the resulting infection risk across the network, which we use as an estimate for both the risk of downstream contamination inside the network and spillover to naïve populations outside of the network.

Here we combine three crucial elements: (1) Economic Decision making in response to contagion, (2) Contagion dynamics in a network, and (3) Assessments of risk from that network. Our model captures the bidirectional coupling of the disease processes and the economic decision making in a way that is robust and flexible enough to be used to describe the qualitative likely behaviors and outcomes of contagion in trade networks. By using a game theoretic framework we capture the economic decision making of the stakeholders, including their responses to infection information. The disease process is captured very simply but the key insight is that, for any acyclic trade network, a stochastic infection process can be summarized with long-time expected infection probability. This expected infection probability gives us a way to calculate best responses and thus find equilibria. While this model does not have the specificity to make exact

quantitative predictions, it does give ecologists, wildlife disease epidemiologists, and economists a way to describe and interrogate qualitative differences among proposed methods for intervention and/or control (e.g., via either standards for hygienic practice or economic incentive) across different network topologies.

Our model focuses on the scale of individual decision-makers within the trade network (as opposed to nations or sectors of the market). Each player's choices regarding trade partner connection and infection management practices are therefore emergent resulting properties of system, ultimately influenced by consumer pressures on economic and epidemiological behaviors. In this way, our model allows exploration into how intervention policies may propagate backwards through trade networks to increase healthy and protective behaviors. For any individual trade network and infection risk, some of the earlier work could be used to parameterize the explicit strategies and payoffs of our game, however our primary goal is not to make specific, quantitative recommendations for any one trade network, but rather to explore the emergent properties of all such systems and whether/how they may be influenced.

The key feature of this model is that it captures the bidirectional coupling of individual choice in the optimization problem with respect to the global setting (i.e., the entire network) and the risk of infection that emerges from the global setting. Individuals make choices with risk of infection in mind, risk of infection changes as a result of these choices, and so the resulting equilibrium is not a product of either one individually but rather a product of their interactions. This means that, in addition to controls on contagion entering and being passed through the system, controls on the payoff structure, which determine individual behavior, are also potent to change risk, even on a global scale.

From the numerical example, we see that changing the payoff structure for subset of players in the model (as in example 5.3) can have a potent effect upstream. This means that our model is able to tease out the impact of a single stakeholders actions. In large systems like trade networks, it is easy to believe that individual actions do not impact overall outcomes measurably but this method of modeling provides a way to describe the consequences of individual actions. In the case of defection, we see that in the toy model one player's choice to defect does little to change the strategies of the others but does greatly change the spillover risk, especially when the defector is far upstream. Although defection is not rational, and a stakeholder acting in their own best interest would not defect, it may be the case that disinformation or strong beliefs may lead stakeholders to defect. This model may lend us to believe that having stakeholders who are well informed about the conditions of the network in which they trade can protect the interests of everyone in the network as well as reduce spillover risk outside the network. In addition, when considering the example of parameter cascades we see that changing incentive structures can have large impacts on the network as a whole. This is of note because, in the application area, it means that should external controls of payoff structures (e.g., subsidies for health and safety measures) be implemented in such a trade network, uniform enforcement in required in order for the controls to have the desired effect.

Although our toy model does little to capture the specific conditions of a real life trade network, it does help us tease apart the mechanisms of upstream decision cascades. When an individual downstream has a change to their payoff function, their best response changes accordingly. When players downstream care more about health practices they force their upstream neighbors to invest more heavily in such practices, not for health's sake but rather driven by profit maximization. The effect is more noticeable when considered in the opposite direction. When consumers care little for health and safety practices, those decisions are readily passed upstream to distributors and producers. Distributors are free to maximize their profits by not investing at all in the health of their stock, so long as the consumers do not care. This may demonstrate the important insight that the upper limit for investment in health and safety upstream is difficult to increase from a downstream position but the floor is almost entirely dependent of the strategies of the downstream stakeholders. Strategies of distributors are limited above by the strategies of the producers upstream, those stakeholders which set the prices at which the distributors must operate. However, the lower limit is set by the stakeholders downstream and their considerations for the health of the products being distributed.

This insight, interestingly, does not entirely extend to the case of defection from a downstream position. When a consumer defects we see that the strategies of the distributors and the producers change very little. This is due to the particular form of the toy model. When a consumer defects to zero, the stakeholders upstream of that consumer get a payoff of exactly $w_{j,i}$ regardless of infection status. Because this is a constant it does not have an impact on the optimization process or the location of the optimum. Likewise, when a consumer defects to one, the stakeholders upstream get no payoff from transacting with the defector and so it does not impact the optimum. In either case, it is the same as if the defector was simply removed from the network. Additionally, because consumers do little to impact the $R_0$ of the network, this defection has almost no measurable impact on the system as a whole. A single defection is dangerous only when it occurs upstream because such upstream effects can lead to increased infection throughout the entire network. It may also be the case that with sufficiently many consumers defecting, the distributors, beginning to be considered terminal stakeholders in the network, may have a qualitative shift in their behavior. This is why we note that a *single* defection is only dangerous when it occurs upstream.

This toy model, of course, does little to tell us about any particular real-world trade network, but it does demonstrate a promising new way to conceptualize and model these systems. In particular, it gives a direct way to make qualitative predictions about spill over risk and intrinsic risk given different control scenarios. Both individual controls and network wide controls can easily be tested through this model and both global and local measures of risk can easily be observed. The model is highly flexible and, provided a sensible tie-breaking ordering is supplied, a best response function can almost always be proven to exist. The optimization [26] is not computationally expensive and so, even with more complicated, nonlinear, functional forms, this model can be used numerically without issue to draw conclusions about multiple measures of risk in and around trade networks.

Of course, were we to attempt to parameterize our model to accurately and precisely reflect any individual real-world wildlife trade network, we could quantify the sensitivity of the relative outcomes for the emergent network structure of trade partnerships/volume and health protective behaviors. However, obtaining these data is itself so complicated as to be prohibitive during these first efforts. Recent work to understand consumers and trade participants beliefs about health practices have begun the costly and painstaking survey work needed to understand economic- and values-based incentives at work [27,28]. Similarly, research has been done to disentangle to structure of the network of organizations which govern the wildlife trade [29,30] which will be crucial in order to turn insights from this model into actionable strategies. Estimating the probability of introduction of infection into a supply chain from the diversity of sources from which animals are provided varies greatly not only by pathogen of concern, but by also by a vast diversity of extrinsic environmental and ecological factors [31,32]. Estimating the impact of hygienic practices faces the challenge not only of careful lab study, but then of understanding how recommended protocols may be enacted by untrained individuals [33]. Each of these elements are individually complicated and costly to characterize well, and we therefore do not focus on any analysis of carefully parameterized scenarios - rather, we focus on the system-wide causes and consequences of frame shifts in expected behaviors and outcomes.

We have demonstrated a new way of capturing the bidirectional coupling of economic decision making and infection dynamics in trade networks of products susceptible to contagion. With the key insight that, as long as the trade network is acyclic, the long time probability of infection is stable and directly computable, we are able to find best responses with a very general set of payoff functions in this framework. With computable best responses and infection probabilities, the model we propose allows us to consider equilibria in trade network systems and measure the associated risk. This means this flexible and robust model is a potent tool to help us answer questions about controls in networks and their effects on infection/spillover risk.

## Supporting information

**S1 Appendix. A necessary elementary analysis lemma.**
(PDF)

## Author contributions

**Conceptualization:** Jesse L. Brunner, Nina H. Fefferman.

**Formal analysis:** John S. McAlister, Nina H. Fefferman.

**Funding acquisition:** Jesse L. Brunner, Nina H. Fefferman.

**Investigation:** John S. McAlister, Danielle J. Galvin, Nina H. Fefferman.

**Methodology:** John S. McAlister.

**Project administration:** Nina H. Fefferman.

**Resources:** Jesse L. Brunner, Nina H. Fefferman.

**Software:** John S. McAlister.

**Supervision:** Nina H. Fefferman.

**Validation:** John S. McAlister.

**Visualization:** John S. McAlister.

**Writing – original draft:** John S. McAlister, Jesse L. Brunner, Danielle J. Galvin, Nina H. Fefferman.

**Writing – review & editing:** John S. McAlister, Jesse L. Brunner, Danielle J. Galvin, Nina H. Fefferman.

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
