## [Decision Letter · Decision Letter 0]

25 Aug 2025

PCOMPBIOL-D-25-01334

A Game Theoretic Treatment of Contagion in Trade Networks

PLOS Computational Biology

Dear Dr. McAlister,

Thank you for submitting your manuscript to PLOS Computational Biology. After careful consideration, we feel that it has merit but does not fully meet PLOS Computational Biology's publication criteria as it currently stands. Therefore, we invite you to submit a revised version of the manuscript that addresses the points raised during the review process.

Please submit your revised manuscript within 60 days Oct 25 2025 11:59PM. If you will need more time than this to complete your revisions, please reply to this message or contact the journal office at ploscompbiol@plos.org. Please include the following items when submitting your revised manuscript:

We look forward to receiving your revised manuscript.

Kind regards,

Christian Hilbe

Academic Editor

PLOS Computational Biology

Benjamin Althouse

Section Editor

PLOS Computational Biology

**Additional Editor Comments :**

The two reviewers both agree that the paper is well motivated, and that in general, the paper is technically sound.

However, Reviewer #2 also argues that the paper is perhaps more abstract than typical articles published in PLoS Computational Biology, and that non-mathematicians will find it difficult to follow this article.

Based on my own reading, I agree with this assessment. For many of the variables introduced in this paper, it would be useful to get some interpretations of what these variables could refer to in the specific application the authors have in mind. As such, the paper seems to be written for a more math-oriented journal, and the authors might want to discuss whether PLoS Computational Biology is really the best fit for their work.

Having said that, both reviewers make a number of very constructive suggestions on how to improve the paper. If the authors can address all of those, the paper may also become suitable for PLoS Computational Biology.

**Journal Requirements:**

At this stage, the following Authors/Authors require contributions: John S. McAlister, Jesse L Brunner, Danielle J Galvin, and Nina H Fefferman. Please ensure that the full contributions of each author are acknowledged in the "Add/Edit/Remove Authors" section of our submission form.

4) Your manuscript is missing the following section: Methods.  Please ensure all required sections are present and in the correct order. Make sure section heading levels are clearly indicated in the manuscript text, and limit sub-sections to 3 heading levels. An outline of the required sections can be consulted in our submission guidelines here:

5) Please upload all the main figures as separate figure files in .tif or .eps format. For more information about figure files please see our guidelines:  

https://journals.plos.org/ploscompbiol/s/figures#loc-file-requirements  

6) The file inventory includes two files for Figures 5. We would recommend either combining these into a single Figure 5.tiff file with separate internal panels, or renumbering them as individual figures, as we are not able to publish multiple components of a single figure as separate files

7) We notice that your supplementary information is included in the manuscript file. Please remove them and upload them with the file type 'Supporting Information'. Please ensure that each Supporting Information file has a legend listed in the manuscript after the references list.

8) Thank you for stating "All the data in this manuscript was synthetically generated by the model using the code found in the repository https://github.com/feffermanlab/730 JSM_2024_WildlifeTradeNetworks." This link reaches a 404 error page. Please update the link provided in the online submission form to "https://github.com/feffermanlab/JSM_2024_WildlifeTradeNetworks."

9) Please amend your detailed Financial Disclosure statement. This is published with the article. It must therefore be completed in full sentences and contain the exact wording you wish to be published.

1) If the funders had no role in your study, please state: "The funders had no role in study design, data collection and analysis, decision to publish, or preparation of the manuscript."

**Reviewers' comments:**

Reviewer's Responses to Questions

**Comments to the Authors:**

**Please note that two reviews are uploaded as attachments.**

Reviewer #1: Please see the uploaded attachment

Reviewer #2: My comments are given in a separate pdf attachment that I have uploaded

**Have the authors made all data and (if applicable) computational code underlying the findings in their manuscript fully available?**

Reviewer #1: Yes

Reviewer #2: Yes

PLOS authors have the option to publish the peer review history of their article (what does this mean?). If published, this will include your full peer review and any attached files.

Reviewer #1: No

Reviewer #2: No

**Figure resubmission:**
---

## [Decision Letter · Decision Letter 1]

11 Dec 2025

PCOMPBIOL-D-25-01334R1

A Game Theoretic Treatment of Contagion in Trade Networks

PLOS Computational Biology

Dear Dr. McAlister,

Thank you for submitting your manuscript to PLOS Computational Biology. After careful consideration, we feel that it has merit but does not fully meet PLOS Computational Biology's publication criteria as it currently stands. Therefore, we invite you to submit a revised version of the manuscript that addresses the points raised during the review process.

We look forward to receiving your revised manuscript.

Kind regards,

Christian Hilbe

Academic Editor

PLOS Computational Biology

Benjamin Althouse

Section Editor

PLOS Computational Biology

**Additional Editor Comments (if provided):**

The manuscript has been evaluated by the same two reviewers who have already handled the original submission. Both reviewers agree that the authors have taken their suggestions into account, and Reviewer #1 suggests acceptance (highlighting a smaller issue that the authors still need to resolve). Reviewer #2 argues that the paper is technically sound and interesting; however, PLoS Computational Biology might not be the optimal venue.

Based on my own reading, I understand this reviewer's point. Articles in PLoS Computational Biology typically have a strong computational component, whereas the strengths of this article lie in its mathematical analysis.

I agree that non-mathematicians will find this article more difficult to understand than mathematicians. But I do believe the authors make a good effort to explain the objects they work with, and that PLoS Computational Biology can serve as a home for articles with a more mathematical audience (if the biological implications are clear, and if computational questions are discussed). Those requirements are met.

Overall, I think that this article still falls within the scope of articles published in PLoS Computational Biology, and hence I lean towards accepting the paper. In this case, I would opt for "Minor revisions", to give the authors a chance to address the remaining minor issues.

**Journal Requirements:**

Please ensure that the funders and grant numbers match between the Financial Disclosure field and the Funding Information tab in your submission form. Note that the funders must be provided in the same order in both places as well. This is published with the article. It must therefore be completed in full sentences and contain the exact wording you wish to be published.

**Reviewers' comments:**

Reviewer's Responses to Questions

**Comments to the Authors:**

Reviewer #1: Minor comment: The naive risk is used as an upper bound for the probability of infection at some node. Of course this is intuitive since we expect infections to be positively correlated rather than negatively correlated, but it's not immediately obvious mathematically why it would be an upper bound. If you don't want to justify this mathematically, I would state it only as a non-rigorous heuristic, or else remove it entirely.

Reviewer #2: The authors have addressed the issues raised by both reviewers and have corrected the mistakes/typos of the previous version. Changes have been made to specifically address the issues raised by me in my previous report. This has led to some improvement, but the structure and presentation style of the paper has remained largely unchanged.

As I mentioned in my first report, the overall message is interesting and the paper seems to be technically sound. However, I am still quite skeptical about the suitability of the paper for PCB. This is not meant to detract from the mathematical results (section 2-3) but is based on my opinion that PCB is not the most appropriate journal for dissemination of these results that are primarily mathematical in nature with minimal application of computational methods. However, I will defer to the editor’s judgement on this matter.

Additional minor comments

1. L130-131: If ‘i’ is the focal player (consumer) and j is the seller, then shouldn’t line 130 read as, “…the benefit of the player ‘j’ is given b_ji…” The statement can perhaps be framed in a different way to enhance clarity….. “The cost of such an interaction for the focal player ‘i’ is given by c_i,j(x_i,x_j) when ‘i’ buys a good from an upstream ‘j’ player; and the benefit for the seller j is given by b_j,i(x_i,x_j).”

2. L248-250 has been written in red and then crossed out. Why?

3. L446-450: I did not understand the meaning of the statement in line 449 that says “..the intrinsic benefit is not measured in a qualitative measure“enjoyment”…….” I thought that the intrinsic benefit is being quantified through the strategy (x_i) of the focal player and her infection status (I_i).

4. L479 should read: “…and the other two buy from both s1 and s2 evenly.”

**Have the authors made all data and (if applicable) computational code underlying the findings in their manuscript fully available?**

Reviewer #1: Yes

Reviewer #2: Yes

PLOS authors have the option to publish the peer review history of their article (what does this mean?). If published, this will include your full peer review and any attached files.

Reviewer #1: No

Reviewer #2: No

**Figure resubmission:**
---

## [Editor Report · Decision Letter 2]

15 Dec 2025

Dear Mr McAlister,

We are pleased to inform you that your manuscript 'A Game Theoretic Treatment of Contagion in Trade Networks' has been provisionally accepted for publication in PLOS Computational Biology.

Best regards,

Christian Hilbe

Academic Editor

PLOS Computational Biology

Benjamin Althouse

Section Editor

PLOS Computational Biology

The authors have addressed all the remaining small remarks satisfactorily.

This is a good paper, and it is appropriate for PLoS Computational Biology.

---

## [Editor Report · Acceptance letter]

PCOMPBIOL-D-25-01334R2

A Game Theoretic Treatment of Contagion in Trade Networks

Dear Dr McAlister,

I am pleased to inform you that your manuscript has been formally accepted for publication in PLOS Computational Biology. Your manuscript is now with our production department and you will be notified of the publication date in due course.

With kind regards,

Narmatha Raju, M.Sc
